# Re Learning Memory Guided Normality for Anomaly Detection

## Reproducibility Summary

**Scope of Reproducibility**

The authors have introduced a novel method for unsupervised anomaly detection that utilises a newly introduced "Memory Module" in their paper [1]. We validate the authors' claim that this helps improve performance by helping the network learn prototypical patterns, and uses the learnt memory to reduce the representation capacity of Convolutional Neural Networks. Further, we validate the efficacy of two losses introduced by the authors, "Separateness Loss" and "Compactness Loss" presented to increase the discriminative power of the memory items and the deeply learned features. We test the efficacy with the help of t-SNE plots of the memory items.

**Methodology**

The authors provide a codebase available at https://github.com/cvlab-yonsei/MNAD with scripts for the Prediction task. We reused the available code to build scripts for the Reconstruction task and variants with and without memory. The completed codebase for all tasks available at https://anonymous.4open.science/r/40136d86-adcb-44cf-8f71-b70ad55a1bde/.

**Results**

We obtain results that are within a 3% range of the reported results for all datasets other than on the ShanghaiTech dataset[2]. The anomalous run stuck out since the "Non Memory" of the same run could score markedly better than the "Memory" variant. This led us to investigate the behaviour of the memory module and found valuable insights which are presented in Section 5.

**What was easy**

- The authors provide codes to run the Prediction task with memory.
- The ideas presented in the paper were clear and easy to understand.
- The datasets were easily available, with the exception of the ShanghaiTech dataset[2].

**What was difficult**

- After training the model on the ShanghaiTech dataset[2], we found that the "Memory" variants of the model had issues as described in Sections 4 and 5. We investigated the behaviour of the memory module and introduced additional supervision to solve the problem.
- We required hyperparameter optimization to hit the reported scores.
- The paper also lacked information as to which parts of the pipeline were actually to be credited for the improvement in performance, as discussed in Section 5.

**Communication with original authors**

The codebase has scripts only for the Prediction task "Memory" variant of the benchmarks. We reused portions of the codebase to create the other variants. We then validated our modifications with the authors. We also communicated the discrepancies we saw in parts of the results and the authors asked us to wait until they update their repository with the missing scripts.

# 1 Introduction

The paper that we have tried to reproduce attempts to detect anomalous events in a video sequence. Anomaly detection based on Convolutional Neural Networks (CNNs) usually leverage proxy tasks like reconstruction of input frames or prediction of future frames, i.e., trained only on the "Normal" scenes thereby having a hard time reconstructing anomalies. Conventionally, Peak Signal-To-Noise ratio (PSNR) is used to gauge if the input batch of images has anomalous frames. The authors have introduced a hybrid metric for Abnormality Score calculation, comprising a weighted sum of the PSNR value and L2 loss between the Queries and the Memory items.

We utilise the term "Reconstruction" to refer to the model and the term "reconstruction" to refer to the output generated by the model (s).

The author's primary contribution is the introduction of a "Memory Module" which is claimed to memorize prototypical patterns of normal data, and these memory items are used to reduce the representation capacity of the CNN which, in theory, decreases the quality of the output (Reconstruction or Prediction) when the input batch has anomalous frames, thereby making it easier to classify input batches as anomaly or not. The authors also introduce two losses, namely, Compactness loss and Separateness loss which help increase the diversity and discriminative power of the memory items. The final contribution is the introduction of Test-Time Updation of memory to help tune the memory items according to the normal scenes during inference.

# 2 Scope of reproducibility

We have attempted to reproduce the reported scores of the "Memory" variant and "Non Memory" variant on all datasets for both the Prediction and Reconstruction tasks, alongside reproducing the scores for the ablation studies as well. The author's central claim is that the "memory module" helps the network learn prototypical patterns of normal scenes, and using the learnt memory items as "noise" reduces the representation capacity of the CNN and makes it harder for the network to reconstruct/predict anomalous output thereby making it increasingly easier to classify anomalous input batch of images.

We ran experiments to compare the performance of the Reconstruction and Prediction models with and without memory modules. While benchmarking these models we observed a few deviations in the reported results and reproduced results which lead us to experiments to get more clarity on the behaviour of the memory module, which we shall discuss in Section 4. We provide the hyperparameters with which we got the best results on the provided datasets.

Their secondary claims revolve around the Compactness loss and Separateness loss, which they claim helps the network space out and the memory items and decreases the spread between similar deeply learned features. They also claim that the Test-Time updation of the memory items helps tune the memory items to the normal scenes during inference time, subsequently classifying anomalies more accurately. We attempted to reproduce the ablation studies results which covers both the secondary claim and the Test-Time updation scheme claim.

The claims we are testing:

- "Memory module" helps the network to learn prototypical patterns of normal scenes, and using learnt memory items to lessen the capacity of CNNs.

- feature compactness and separateness losses to train the memory, ensuring the diversity and discriminative power of the memory items. improves performance.

- update scheme of the memory, when both normal and abnormal samples exist at test time, improves performance.

# 3 Methodology

## 3.1 Model descriptions

The authors present a novel algorithm for unsupervised anomaly detection that makes use of a memory module. The model architecture comprises of three parts: The encoder, the memory module and the decoder. The model either reconstructs or predicts a video frame with a combination of features from the encoder and the memory items rather than just the features from the encoder.

The memory module consists of 10 items of 512 each. A "read" operation generates a tensor of size 32*32*512 from the memory module which is then concatenated to the tensor obtained from the encoder. This obtained matrix is

concatenated with the query features to generate a tensor of depth 1024, then passed to the decoder. The key idea here is that the memory item is used to generate a tensor using a weighted average, thus utilising all the memory items. This allows the model to learn diverse patterns. The network and memory module is trained only on normal frames. As a result, the cosine distance between the query and the relevant memory item is less and they are close to each other. On the contrary, since the network has not seen the anomalous frames during the training phase, the memory item will be far from the queries obtained. Since they are far, the Query-Memory pair will be sub-optimal thus resulting in poor reconstruction of the input frames.

To read the items, the cosine similarity score is computed between each query $q_t^k$ and all memory items $p_m$, resulting in a 2-dimensional correlation map of size $M \times K$, A softmax function is then applied along a vertical direction, and matching probabilities $w_t^{k,m}$ are obtained as shown in Equation 1.

$$w_t^{k,m} = \frac{\exp\left((p_m)^T q_t^k\right)}{\sum_{m'=1}^{M} \exp\left((p_{m'})^T q_t^k\right)}, \tag{1}$$

For each query $q_t^k$, the memory items are read by a weighted average of the items $p_m$ with corresponding weights $w_t^{k,m}$ and obtain $\hat{p}_t^k$ as shown in Equation 2. The $\hat{p}_t^k$ values computed from each query makes up the feature map which is then concatenated to the features obtained from the Encoder depth wise.

$$\hat{p}_t^k = \sum_{m'=1}^{M} w_t^{k,m'} p_{m'}, \tag{2}$$

The "update" operation updates the memory items as it iterates through the dataset and learns the pattern of normal features. To update the memory, for each item, all queries that are closest to the memory item are chosen using the matching probabilities in Equation 1. The set of indices for the corresponding queries for $m$-th item in the memory are denoted by $U_t^m$. The item using the queries indexed by $U_t^m$ is updated as shown in Equation 3. $f(\cdot)$ is the L2 norm.

$$p^m \leftarrow f\left(p^m + \sum_{k \in U_t^m} {v'}_t^{k,m} q_t^k\right), \tag{3}$$

A weighted average of the queries is used. The matching probabilities are computed by applying the softmax function to the correlation map of $M \times K$ along a horizontal direction as shown in Equation 4.

$$v_t^{k,m} = \frac{\exp\left((p_m)^T q_t^k\right)}{\sum_{k'=1}^{K} \exp\left((p_m)^T q_t^{k'}\right)} \tag{4}$$

The authors choose to use PSNR as a part of the metric to check the output quality from the decoder. Along with this, the authors utilise an L2 distance loss calculated between the queries and its corresponding nearest memory item. The PSNR and L2 distance is coupled together as shown in Equation 5.

$$S_t = \lambda \left(1 - g\left(P\left(\hat{I}_t, I_t\right)\right)\right) + (1 - \lambda) g\left(D\left(q_t, p\right)\right), \tag{5}$$

The authors also introduce a new pair of losses, Compactness loss and Separateness loss. The Separateness loss minimizes the distance between each feature and its nearest item, while maximising the discrepancy between the feature and the second nearest one, similar to Triplet loss [3]. The feature Compactness loss maps the features of a normal video frame to the nearest item in the memory and forces them to be close. These losses coupled, spread the individual items in the memory and as a result enhance the discriminative power of the features and the memory items.

The architecture used in Reconstruction task "Memory" variant is similar to that of the Prediction task "Memory" variant with the exception that it does not have skip connections. The Prediction "Non Memory" variant resemble a U-Net [4] and the Reconstruction models resemble a simple Autoencoder [5].

The authors have provided a codebase available at https://github.com/cvlab-yonsei/MNAD. However, the codebase has only the codes for the "Memory" variant of the Prediction task[1]. We reused a major portion of the available code

---

[1]Post Submission for review, the authors updated their GitHub repository to include the scripts for the "Memory" variant of the Reconstruction task. Discrepancies in this code are available on our code repository.

to build scripts for the Reconstruction task and variants with and without memory. We validated the new scripts and changes by communicating with the authors over email to ensure there is no gap between their intention and our developed codebase. The completed codebase for all tasks available at https://bit.ly/3s3nTIR. The available code for the "Prediction" task could be run on the Ped2[6] and CUHK[7] datasets without any changes. However, the ShanghaiTech dataset[2] required us to generate the frames from the provided videos. We utilise the VideoCapture class available on the OpenCV library to generate 2,74,515 frames.

We make the following changes in the available code to create the scripts for the Reconstruction task:

- We tweak the code to accept one input and generate one output of the same frame by changing the time step parameter.
- We remove the skip connections present in the architecture for the Prediction task.

We make the following changes for the "Non Memory" variants:

- The decoder needs only the query features and hence we change the first layer of the decoder to accept and create a 512 size vector.
- For the "Non Memory" variants of the models, we cannot calculate the L2 distance and only the PSNR value is to be used for Abnormality Score calculation in the evaluation script.

## 3.2 Datasets

We test and report results on all datasets that the authors have reported scores on. The three datasets used are UCSD Ped2[6], CUHK Avenue[7] and the ShanghaiTech[2] anomaly detection dataset. All three datasets are readily available in the author's repository. However, only the UCSD Ped2 dataset[6] and CUHK Avenue dataset [7] are available directly in the format required by the authors' code. The ShanghaiTech dataset[2] contains only videos which we had to split into frames using OpenCV to generate 274,515 frames. We have provided the script used to save the frames on the GitHub repository.

## 3.3 Hyperparameters

We experimented heavily with the $\lambda$ hyperparameter to achieve the reported scores. We could not find a fixed trend in its behaviour across the three datasets. We have provided the best results after tweaking $\lambda$ and the results obtained as per the author's recommendation of hyperparameters as discussed in Section 4.1. As for the other hyperparameters, we found that the recommended values worked well and gave us similar results to the paper with the exception of the ShanghaiTech dataset[2].

## 3.4 Experimental setup

We use multiple hardware set-ups for our experiments, namely, vast.ai instances, Google Colab and our personal systems. We used NVidia Tesla T4 and P100 from Google Colab, along with 1080 Ti, 2070, 2070 SUPER, 2080 Ti rented on vast.ai. To maintain uniformity, we benchmark all our scores on a 2080 Ti. We recorded all our training metrics on "wandb"[2].

## 3.5 Computational requirements

For our experiments and benchmarking, we use a rented GPU from vast.ai. All our scores are reported on a 2080Ti with 8 GB vRAM.

The training times are as shown in Table 1.
The inference times and model parameters for the Prediction task "Memory" variant and Reconstruction task "Memory" variant are as shown in Table 2 and Table 3 respectively.

# 4 Results

The paper has two major tasks, "Prediction" and "Reconstruction", each of which has a "Memory" and "Non Memory" variant. We test these 4 models on all three datasets that the authors have evaluated on, namely Ped2[6], CUHK Avenue

---

[2]wandb logs are available in the repository.
[3]Denotes the Model size and Memory tensor size

| Model | Training Time (in minutes) | | |
|---|---|---|---|
| | Ped2[6] | CUHK [7] | ShanghaiTech[2] |
| Prediction w Memory | 82 | 562 | 1498 |
| Prediction w/o Memory | 76 | 406 | 1390 |
| Reconstruction w Memory | 67 | 405 | 1207 |
| Reconstruction w/o Memory | 61 | 364 | 1086 |

Table 1: Model Training times in minutes on 2080Ti

| Parameter Category | Parameter | Value |
|---|---|---|
| Storage | Saved Weights Size | 62.7 MB + 20.8 kB [3] |
| Time (GPU) | Inference | 0.012s |

Table 2: Inference times and Memory requirements

[7] and ShanghaiTech[2]. We obtain results that are within a 3% range of the reported results for all datasets other than on the ShanghaiTech dataset[2]. However, we observed that hyperparameter optimization has to be done to obtain the best results possible. The weighted average of the PSNR and L2 distance between query and memory item, $\lambda$ in Equation 5, was the primary hyperparameter we had to tune for optimal results. We have reported the results obtained on the hyperparameter values chosen by the authors and the best obtained results, post hyperparameter optimisation, discussed in Section 4.1.

We further validate the efficacy of the introduced Separateness Loss, Compactness Loss and Test-Time updation that the authors claim improves performance. These results are discussed in Section 4.2. We find that the obtained scores do not follow the trend shown in the original paper.

In our experiments, we found that the results majorly supported claim 1. The t-SNE plots also show that the supervision proposed by the authors work well for ensuring diversity and increasing the discriminative power of the memory items, i.e., supporting the second claim. The results we observed for complex datasets like ShanghaiTech[2] clearly do not support the idea behind Test-Time Updation.

Training the proposed Prediction task "Memory" variant on the ShanghaiTech dataset[2] posed problems, which other developers have also faced (https://github.com/cvlab-yonsei/MNAD/issues/6). The "Memory" variants also performed worse than the "Non-Memory" variants on this dataset. This led us to study the behaviour of the memory module. We checked the distribution of the memory items vs deeply learned features and observed that the distribution was completely lopsided and only a part of the memory items were being used. This was consistent throughout the datasets but the skew in the distribution of the memory items linked to the deeply learned features in the ShanghaiTech dataset[2] was extreme, specifically, only one memory item was linked to all the features. Ped2[6] and CUHK Avenue[7] had 6 and 9 memory items linked to the deeply learned features, respectively. We introduced a new supervision (discussed in Section 5) to force a uniform distribution on the memory distribution to help with this problem and we were able to reproduce the reported scores on the "Memory" variants for both the Prediction and Reconstruction tasks on the ShanghaiTech dataset[2], which previously failed. Our proposed solution is not ideal, but was enough to fix these issues to be able to reproduce the reported scores.

## 4.1 Benchmarking Results

The results obtained on the hyperparameter values provided by the authors are provided in Table 4.

The best obtained results are provided along with the hyperparameter values in Table 5. We evaluated the models saved at every 5th epoch and tweaked values of $\lambda$ and Batch Size.

## 4.2 Ablation Results

In our ablation study experiments, we validate the following:

- We test the efficacy of the Separateness Loss, Compactness Loss and Test-Time updation for both the Reconstruction and Prediction tasks on the UCSD Ped2 dataset[6]. The results are as shown in Table 6 and Table 7.

---

[4]value of $\lambda$ used is 1 due to issues discussed in Section 5.

| Parameter Category | Parameter | Value |
|---|---|---|
| Storage | Saved Weights Size | 42.7 MB + 20.8 kB |
| Time (GPU) | Inference | 0.010s |

Table 3: Inference times and Memory requirements

| | Ped2[6] | CUHK[7] | ShanghaiTech[2] |
|---|---|---|---|
| Prediction w Memory | 96.33% | 87.91% | 67.81% [4] |
| Prediction w/o Memory | 95.09% | 84.58% | 67.9% |
| Reconstruction w/ Memory | 87.34% | 83.08% | 64.14% |
| Reconstruction w/o Memory | 88.53% | 81.06% | 67.72% |

Table 4: Benchmarking results on recommended hyperparameter values

- We compare the distribution of query features, learnt with and without the Separateness Loss on the UCSD Ped2 dataset[6] as shown in Figure 1.

# 5    Additional experiments and inferences

- **Reconstruction with Skip-Connections**: We tried the Prediction model without skip connection and noticed a drop in performance. To check if skip connections were to be credited for the performance boost, we tried to introduce skip connections to the Reconstruction task. The problem with adding skip connections to the Reconstruction task is that, since the input and output are the same, the network quickly learns to copy the input to the output which makes the reconstruction near perfect for any input, be it normal or anomalous. To prevent the network from copying the input, we added salt and pepper noise to the input and the output was the denoised input. This allowed us to train models on the Reconstruction task with skip connections and we straight away saw a performance gain of around 4%. The results from this model do not outperform the results from the Prediction task, but it is important to note that the Reconstruction task only accounts for Spatial anomalies and not Spatio-Temporal or Temporal anomalies. This clearly shows that the skip connections are to be credited heavily for how well the Prediction models perform. Along with skip connections, the model also gets Temporal as well as Spatial information for the Prediction task. To add to all these advantages, in the Prediction task, the input is a batch of 4 consecutive frames and the output is the 5th frame. Since the output is not a subset of the input, this facilitates the use of skip connections without having to worry about the network copying the input. The input to output translation is also more consistent than the salt and pepper noise we used in the "Reconstruction with skip connections" experiments.

  We conclude that the skip connections in the Prediction model are to be credited more than the "frame-prediction" task itself. A more ideal method of benchmarking would be to instead use the difference in architecture, that is the skip connections vs no-skip connections and with Temporal information vs without Temporal information rather than just the task itself, i.e Prediction vs Reconstruction.

  25% SALT and PEPPER noise gave us a best result of 92.23% AUC, a near 4% jump on the Reconstruction task.

| Dataset | Model | $\lambda$ | Epoch | Batch Size | AUC (Ours) | AUC (Original) |
|---|---|---|---|---|---|---|
| Ped2[6] | Pred w/ Mem | 0.52 | 45 | 2 | 97.06% | 97.0% |
| | Pred w/o Mem | – | 55 | 4 | 95.11% | 94.3% |
| | Recon w/ Mem | 0.9 | 50 | 4 | 88.36% | 90.2% |
| | Recon w/o Mem | – | 60 | 4 | 88.53% | 86.4% |
| CUHK Avenue[7] | Pred w/ Mem | 0.7 | 60 | 4 | 87.91% | 88.5% |
| | Pred w/o Mem | – | 60 | 4 | 84.58% | 84.5% |
| | Recon w/ Mem | 0.7 | 40 | 4 | 83.13% | 82.8% |
| | Recon w/o Mem | – | 50 | 4 | 81.20% | 80.6% |
| ShanghaiTech[2] | Pred w/ Mem | 1.0 | 1 | 4 | 70.92% | 70.5% |
| | Pred w/o Mem | – | 10 | 4 | 67.9% | 66.8% |
| | Recon w/ Mem | 0.7 | 10 | 4 | 64.14% | 69.8% |
| | Recon w/o Mem | – | 5 | 4 | 68.47% | 65.8% |

Table 5: Best results obtained after hyperparameter optimization

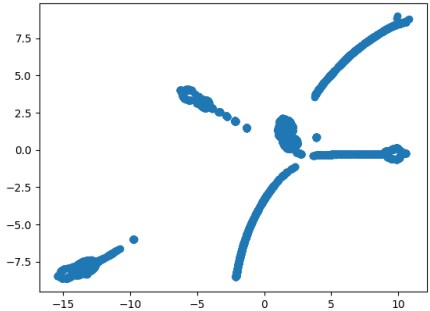
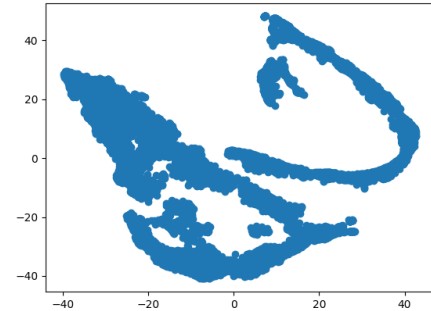

(a) t-SNE plot with Separateness Loss

(b) t-SNE plot without Separateness Loss

Figure 1: t-SNE plots of Query Features on Ped2[6] with and without Separateness Loss

| Separateness Loss | Compactness Loss | Test Time Updation | $\lambda$ | AUC |
|---|---|---|---|---|
| N | N | Y | 0.87 | 84.03% |
| N | Y | Y | 1 | 88.99% |
| Y | N | Y | 0.8 | 87.24% |
| Y | Y | N | 0.8 | 87.72% |
| Y | Y | Y | 0.7 | 87.34% |

Table 6: Ablation results with and without Separateness Loss, Compactness Loss and Test-Time updation for Reconstruction

- **Memory Distribution plots for better understanding the utilisation of memory**: The model architecture offers 10 memory items of size 512 each. During our evaluation of the ShanghaiTech dataset[2], we found that the Abnormality Score vector had a large number of NaNs, due to which we could not evaluate the model. We further investigated the Prediction "Memory" variant of the model in order to fix this bottleneck. To validate our doubts and concerns, we plotted the distribution of the memory items vs the deeply learned features and observed that the distribution was lopsided and only some of the items were being utilised.

  We saw this behaviour repeat across the three datasets however the worst affected was ShanghaiTech[2] where only one memory item was being utilised (See Figure 4a). Ped2[6] and CUHK[7] utilised 6 and 9 memory items respectively but the distribution was still lopsided as it seems that some memory items are utilised more than the others as shown in Figure 2.

  The abnormality score is a proportional summation of the PSNR and L2 distance between the Deeply learned feature and the closest memory item. In case of ShanghaiTech[2], one memory item was responsible for all of the features and due to min-max normalisation used in the abnormality score, the minimum and maximum L2 would be computed to the exact same value which meant that the normalization process would have to divide by 0, which resulted in NaNs. We found that other developers had similar issues as shown in this issue. https://github.com/cvlab-yonsei/MNAD/issues/6

  This problem resulted in Evaluation failing on the ShanghaiTech dataset[2] and returning NaNs in the final anomaly score. A temporary work around was setting the $\lambda$ value to 1, as a result weighting only the reconstruction or PSNR scores and not using the L2 loss between the query and memory items for the Abnormality Score calculation.

  As a solution, we added a new supervision to force a uniform distribution across the memory items. We were able to reproduce the reported results on ShanghaiTech[2] without any issues, which previously failed. The methodology used and results are further discussed ahead.

- **Proposed memory distribution Supervision**:
  We introduce a new supervision to ensure that all the memory items are utilised uniformly. We create a uniformly distributed correlation map of size $10 \times 1024$ which is used to supervise the distribution of Query features across the memory items using Mean Squared Error (MSE) Loss as shown in following the code snippet.

```
loss_mse = torch.nn.MSELoss()
softmax_score_query, softmax_score_memory = self.get_score(keys, query)
```

| Separateness Loss | Compactness Loss | Test Time Updation | $\lambda$ | AUC |
|---|---|---|---|---|
| N | N | Y | 0.7 | 95.70% |
| N | Y | Y | 0.9 | 94.78% |
| Y | N | Y | 0.9 | 95.02% |
| Y | Y | N | 0.6 | 96.96% |
| Y | Y | Y | 1 | 95.13% |

Table 7: Ablation results with and without Separateness Loss, Compactness Loss and Test-time updation for Prediction

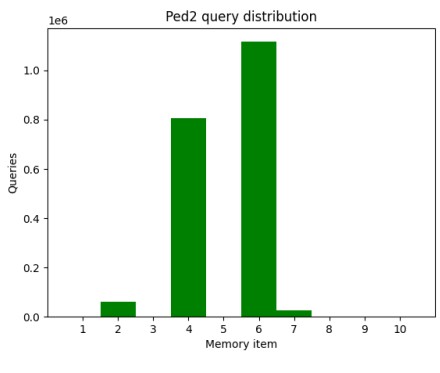

(a) Ped2[6] Query distribution

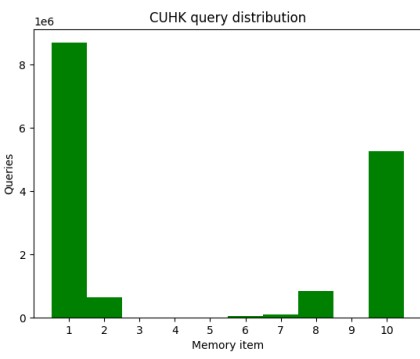

(b) CUHK[7] Query distribution

Figure 2: Query-Memory Distribution histograms across Memory items for Ped2[6] and CUHK[7] datasets

```
244    mem_dist = torch.argmax(softmax_score_memory,dim = 1)
245    mem_dist = F.one_hot(mem_dist, num_classes = 10)
246    mem_dist = torch.sum(mem_dist, dim = 0)
247    query_reshape = query.contiguous().view(batch_size*h*w, dims)
248    mem_dist_loss = loss_mse(softmax_score_memory, self.ideal.float())
```

249  Figure 3 is **only** a representation of the Proposed Memory distribution supervision. This gives a more uniform
250  distribution of query features across the memory items as shown in Figure 4b.

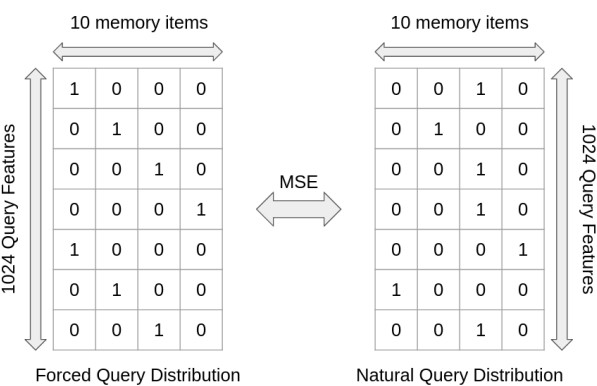

Figure 3: Proposed Memory Distribution supervision

251  • **t-SNE plots to check clusters**:

252  We ran t-SNE plots on the query features obtained from the encoder for the Ped2[6] and CUHK[7] datasets to
253  validate the reported ablation study (Section 4.3 in the original paper). Further, we also obtain t-SNE plots
254  on our introduced supervision. We also compare these with the t-SNE plots obtained from the model trained
255  without Separateness loss discussed in Section 4.2.

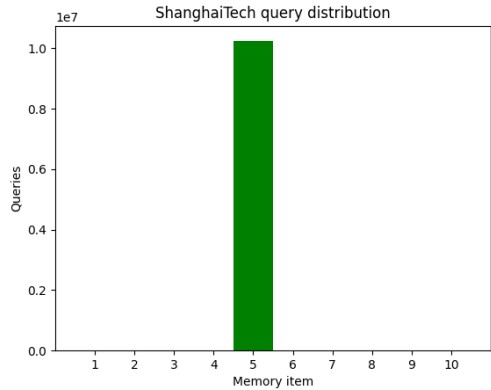
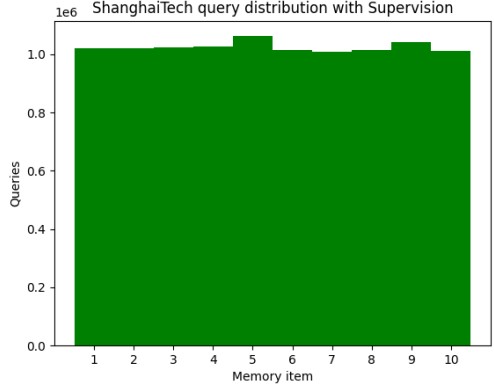

(a) ShanghaiTech[2] Query Distribution

(b) ShanghaiTech[2] Query Distribution with Supervision

Figure 4: Query Memory Distribution across Memory items for ShanghaiTech[2] without and with proposed Supervision

The t-SNE plots for Ped2[6] are shown in Figure 5. We observe that the model trained with the settings provided by the author has 6 and 9 clusters for the Ped2[6] and CUHK[7] datasets respectively. These are equal to the number of memory items utilised by the model.

We also observe that the clusters are spaced out and are thus discriminative with Separateness Loss.

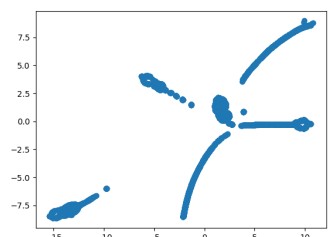
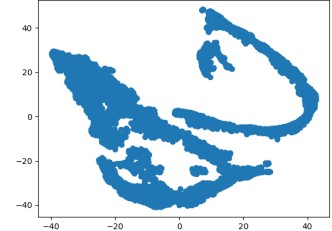
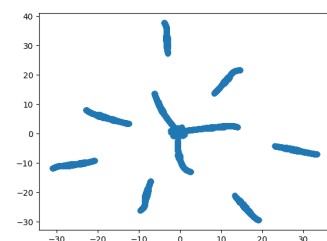

(a) t-SNE plot of Queries with Separateness Loss

(b) t-SNE plot of Queries with no Separateness Loss

(c) t-SNE plot of Queries with proposed supervision

Figure 5: t-SNE plots of Ped2[6] queries with Separateness Loss, without Separateness Loss and Proposed Supervision

- **Results on our newly introduced supervision**: We trained the "Memory" variant of the Prediction and Reconstruction tasks again using our proposed supervision. We were able to evaluate the trained models on the ShanghaiTech dataset[2] with a value of $\lambda$ lesser than 1, which was previously not possible. We were able to beat the scores reported by the authors on both the Prediction and Reconstruction tasks. The obtained results are as shown in Table 8.

  We observed that our obtained scores on the Reconstruction task was more than the Prediction task. The fact that the Reconstruction model performs better than the Prediction model here is deeply concerning as the Prediction model also has Temporal information which helps recognise Temporal and Spatio-Temporal anomalies while the Reconstruction model does not. Thus, the Reconstruction model completely misses out on Temporal anomalies but is still able to outperform the Prediction model. A possible explanation for the same could be the fact that the ShanghaiTech dataset[2] is too complex for the memory mechanism to be utilised as intended. However, we can say with surety that the "Memory" module does not seem to be helping the performance of the model on the ShanghaiTech dataset[2].

  While forcing the model to use each memory item equally is not an ideal solution, it helps the model work without errors and shows the need for a better and dynamic learning scheme for the memory items.

- **Results with and without Test-Time updation**: The authors claim that Test-Time updation of the memory helps improve the performance of the model. However, we found contrary results for the Reconstruction with Memory task on ShanghaiTech[2] and Ped2[6]. For the Prediction task, we empirically found that the 1st skip connection is heavily weighted and is highly responsible for the reconstruction quality. As a result, the

| Model | AUC |
|---|---|
| Reconstruction w/ Memory w/o Proposed Supervision | 64.14% |
| Reconstruction w/ Memory w/ Proposed Supervision | 71.07% |
| Prediction w/ Memory w/o Proposed Supervision | 67.81% |
| Prediction w/ Memory w/ Proposed Supervision | 70.40% |

Table 8: AUC scores on Memory models with Proposed Supervision on ShanghaiTech[2]

intermediate and memory items are not involved in the Prediction task. We passed zeros, ones and random tensors for skip 2,3 and the deeply learnt features from the encoder and the output was optically unchanged for this experiment. As a result, the memory does not seem to affect the output quality much for complex datasets like ShanghaiTech[2]. For Reconstruction task, we can clearly see the drop in performance "with Test-Time-Updation" in ShanghaiTech[2] as there are no skip connections to overcompensate. The results on ShanghaiTech[2] are as shown in Table 9.

| Model | AUC |
|---|---|
| Reconstruction w/ Memory w/ Proposed Supervision w/o Test-Time Updation | 71.07% |
| Reconstruction w/ Memory w/o Proposed Supervision w/ Test-Time Updation | 64.14% |
| Prediction w/ Memory w/ Proposed Supervision w/o Test-Time Updation | 70.40% |
| Prediction w/ Memory w/ Proposed Supervision w/ Test-Time Updation | 70.35% |

Table 9: AUC scores with and without Test-Time updation on ShanghaiTech[2]

- **Synthetic Dataset for Reconstruction vs Prediction**: The Reconstruction task, by virtue of the fact that it takes in only one input and reconstructs one output, is able to only identify only Spatial anomalies. The Prediction task reigns superior in this regard and identifies Spatial, Temporal and Spatio-Temporal anomalies. To check and visualise the reconstruction quality of Spatial and Temporal anomalies, we created a Synthetic dataset comprising Spatial, Temporal and Spatio-Temporal anomalies as shown below.

  We create a Synthetic dataset where the normal instances include circles of radius 10 pixels moving across at a speed of 5 pixels per frame. Spatial anomalies involve squares moving at the same speed while Temporal anomalies involve circles moving at a speed of 10 pixels per frame. Spatio-Temporal anomalies are squares moving at the faster speed.

  We clearly see in Figure 6 that both Reconstruction and Prediction models can reconstruct the normal frames perfectly. Figure 7 shows both Reconstruction and Prediction models have trouble reconstructing the input frame, making it possible to classify these frames as anomalous. Figure 8 shows how the Reconstruction model can easily reconstruct the Temporal anomalies whereas the Prediction model reconstructs these frames with artifacts. Only the Prediction model's outputs would allow for classifying these frames as anomalies, and the Reconstruction model's outputs would classify these as normal frames. Finally, Figure 9 shows both models having trouble reconstructing the input frames, but the Prediction model reconstructs the image with considerably more artifacts making it easier to classify correctly. We use Mean Squared Error (MSE) as a metric to measure the reconstruction quality for the Synthetic Dataset. The MSE value has been provided along with the reconstruction itself in Figures 6, 7, 8 and 9. The MSE has been calculated between the reconstruction and the input frame.

  We observe that the Prediction model sees a marked difference in reconstruction quality for entities both faster, slower and stationary objects where the normal scenario is an object moving at a constant speed.

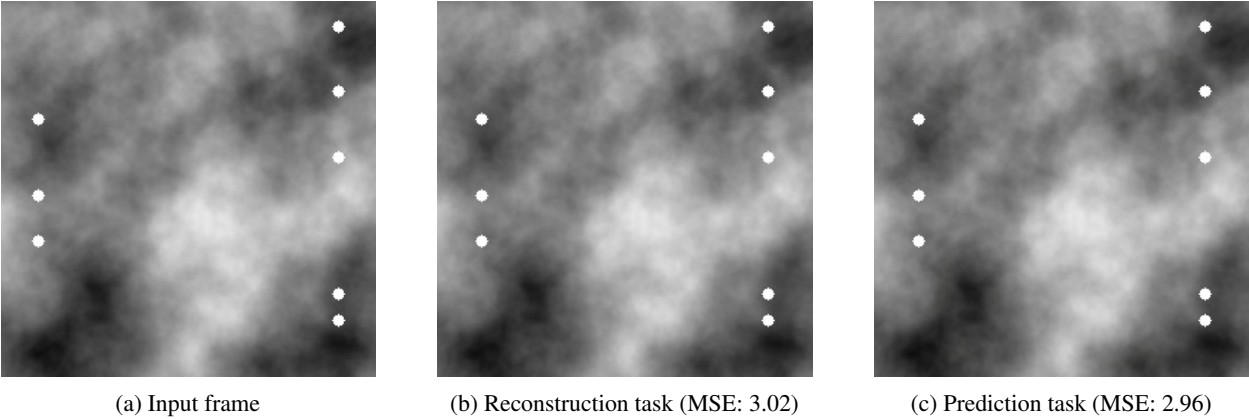

| (a) Input frame | (b) Reconstruction task (MSE: 3.02) | (c) Prediction task (MSE: 2.96) |

Figure 6: Reconstructions of normal frames from Reconstruction and Prediction tasks

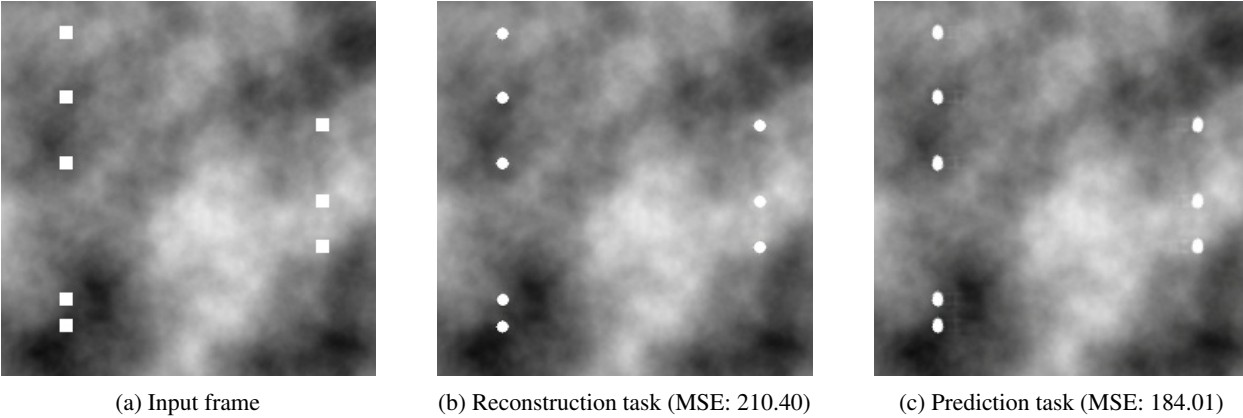

| (a) Input frame | (b) Reconstruction task (MSE: 210.40) | (c) Prediction task (MSE: 184.01) |

Figure 7: Reconstructions of Spatial anomalies from Reconstruction and Prediction tasks

## 6 Discussion

Our results as shown in Section 4.1, show that the memory models do perform better than the ones without for Ped2[6] and CUHK[7] datasets. However, the "Memory" variants achieve lower AUC scores than the "Non-Memory" variants on the ShanghaiTech dataset[2] (Table 4). Our Ablation study experiments on Separateness Loss, Compactness Loss and Test-Time updation do not follow the same trend as the authors, shown in Table 6 and Table 7. However, our generated t-SNE plots (Figure 5) show that the Separateness Loss and Compactness Loss help increase the discriminative power of the Memory items by spacing the Query Features out. Our results in Table 9 show that Test-Time updation fails on complex datasets like ShanghaiTech[2].

As shown in Section 5, the memory distribution is lopsided. Utilising only one memory item results in similar reconstruction for both anomalous and normal frames. We provide a temporary solution to ensure that the model works smoothly. Our newly introduced Supervision is described in Section 5 and is backed with results (Table 8) and histograms to show the uniform distribution of features among the memory items as shown in Figure 4b .

While most of our results corroborate with the claims made by the authors, we believe there are a number of changes that can be brought about to improve performance. As shown in Figure 4a, in Section 5, for a complex dataset like ShanghaiTech[2], only one memory item is being utilised. This is against the core idea of the paper, that is to reduce the representative power of CNNs.

The utilisation of just one memory item means that irrespective of the features being close or far, the same tensor obtained after reading the memory is concatenated to the features obtained from the encoder. As a result, there is no difference in reconstruction quality for anomalous features as compared to the normal features. In Figure 10, we illustrate how a number of query features correspond to a particular Memory item and how it is concatenated depending on the cosine distance from the query features.

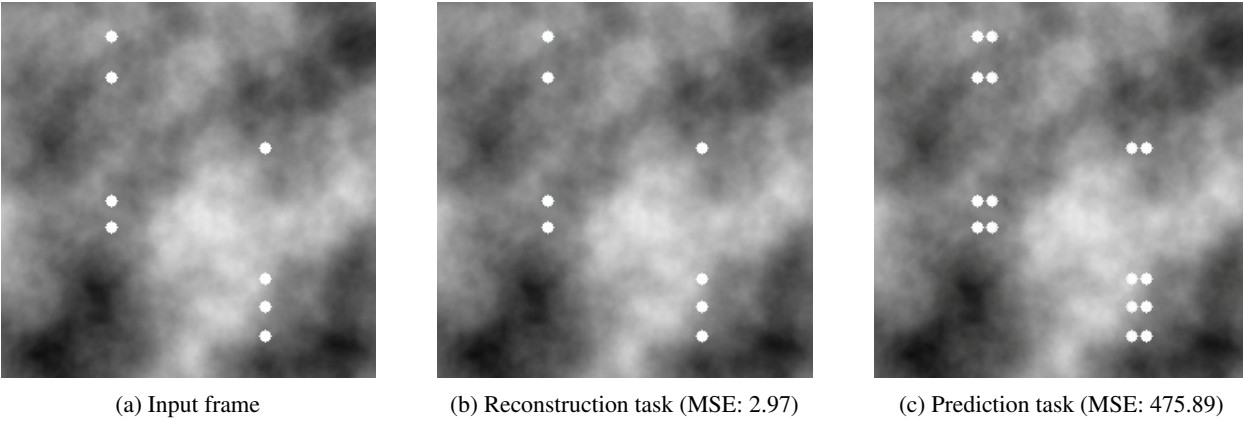

| (a) Input frame | (b) Reconstruction task (MSE: 2.97) | (c) Prediction task (MSE: 475.89) |

Figure 8: Reconstructions of Temporal anomalies from Reconstruction and Prediction tasks

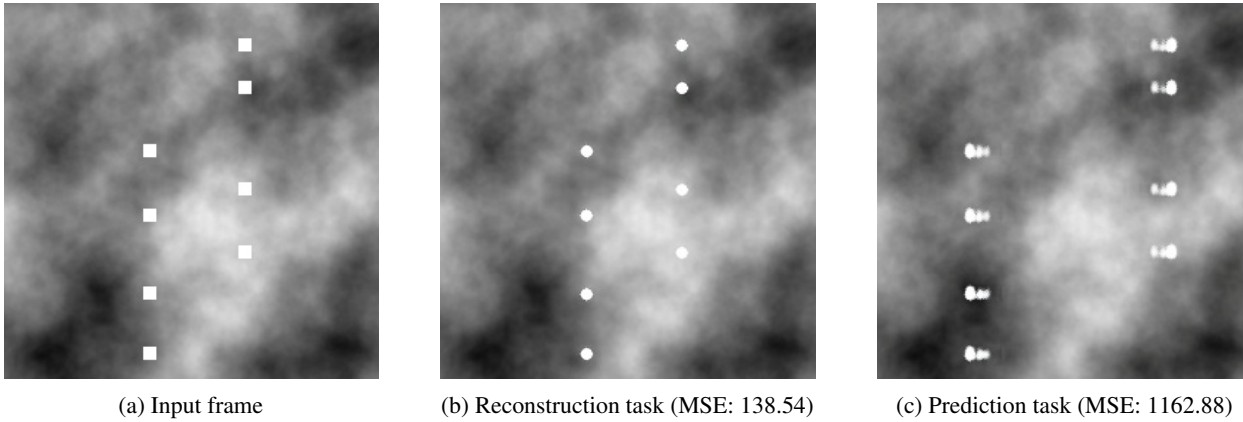

| (a) Input frame | (b) Reconstruction task (MSE: 138.54) | (c) Prediction task (MSE: 1162.88) |

Figure 9: Reconstructions of Spatio-Temporal anomalies from Reconstruction and Prediction tasks

While our proposed supervision is not ideal, we were able to get this script running and were able to achieve the reported scores by the author as shown in Table 8 without any issues described in Section 5.

Further, for ShanghaiTech[2], we found that the Reconstruction models with our new Supervision seem to work better than the Prediction with Memory models. Given the fact that the Prediction task has Temporal information as well, it should be outperforming the Reconstruction task due to the correct classification of Temporal anomalies. Thus, the Reconstruction task outperforming the Prediction task shows that the model does not work as intended.

Finally, we show the difference between the reconstruction of Spatial, Temporal and Spatio-Temporal anomalies from both the Prediction and Reconstruction models as depicted using the Synthetic Dataset discussed in Section 5 and as shown in Figures 6, 7, 8 and 9.

## 6.1 What was easy

- Training and evaluation of the Prediction task "Memory" variant is straightforward with the codebase that the authors provide.

- The codebase was clear enough for us to easily reuse it for implementing the missing variants discussed in the paper, namely the Prediction "Non Memory" variant and both the "Memory" and "Non Memory" variants of the Reconstruction task.

- The ideas presented in the paper were clear and easy to understand.

- All benchmarking datasets were readily available in the format required by the codebase, with the exception of the ShanghaiTech dataset[2].

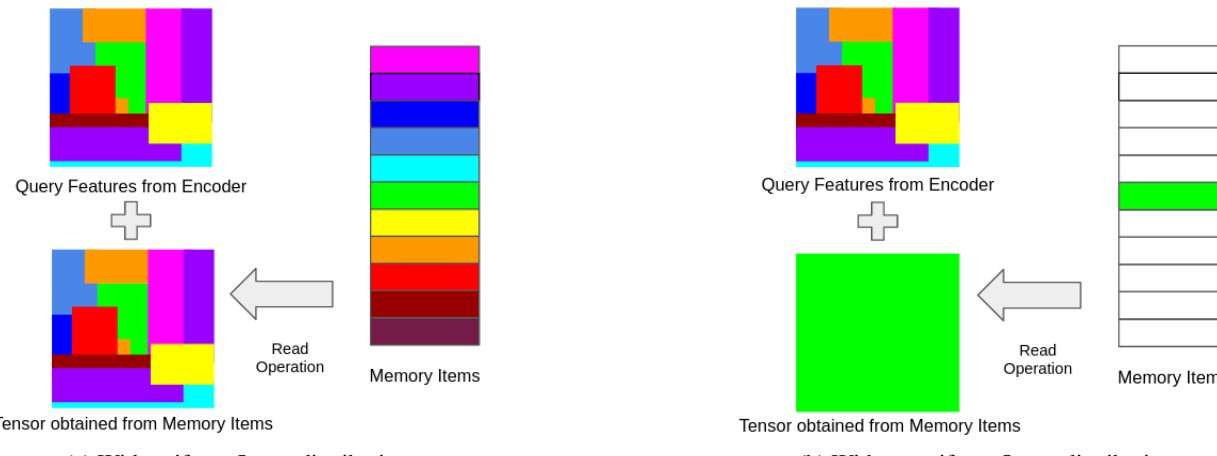

|(a) With uniform Query distribution|(b) Without uniform Query distribution|

Figure 10: Concatenated Tensor with and without Uniform Query Distribution

### 6.2 What was difficult

- Training on the ShanghaiTech dataset[2] took 25 hrs, and once trained, we observed the "Memory" variants had clear issues (discussed in Section 5) which made it difficult to hit the reported scores. We further investigated the behaviour of the memory module and had to introduce additional supervision to help with the problems (Section 5).

- We found that the hyperparameters specified by the author were not suitable for a number of tasks and thus needed hyperparameter optimization to hit the reported scores.

- There was also a want of information on which parts of the pipeline were to be credited for the results. The heavy emphasis on benchmarking on the Reconstruction vs Prediction tasks misdirects the reader from why certain inclusions and modifications in the network architecture is the reason for reported state of the art performance (discussed in Section 5).

### 6.3 Communication with original authors

The codebase has scripts only for the Prediction task "Memory" variant of the benchmarks. We reused the majority of the codebase to create the other variants so as to have minimum variation from what the authors intended. We then validated our modifications with the authors through e-mail communication and will have all correspondence available in the code repository. We also communicated the discrepancies we saw in parts of the results and the authors told us they would update their repository with the missing scripts soon to help match the reported scores.

## 7 Reproducibility Recommendations

We have the following recommendations:

- We believe that the ideal value of the hyperparameter $\lambda$ is 0.7-0.8 for most datasets as this seems to yield the best results for the "Memory" variant of the Prediction task as compared to 0.6 suggested by the authors.

- There also appears to be a problem with PyTorch versioning where the best results are obtained with PyTorch 1.1.0. This is an issue that other developers noticed as well (https://github.com/cvlab-yonsei/MNAD/issues/1). The reason for this behaviour is unclear.

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
