# OpenReview forum: "Re Learning Memory Guided Normality for Anomaly Detection"
_ML_Reproducibility_Challenge/2020 — RC2020_

### Official Review · AnonReviewer1 · 2021-03-01
**Excellent reproducibility report, but implications of findings slightly unclear**

**Rating:** 9
**Confidence:** 4

**Review:**

I would like to congratulate the authors on their commendable effort in preparing this reproducibility report. I believe the authors' effort will significantly contribute to the improvement of the reproducibility of this work. I have presented a more detailed evaluation below.

__Pros:__
1. The authors did a great job in presenting the reproducibility summary, especially the easy and hard parts of their reproducibility effort.
2. The introduction and scope of reproducibility are clearly presented and nicely setup the rest of the report. I would also like to appreciate the authors' effort to present the mathematical foundations of the method. However, I feel the grammar and writing style could be improved for better readability.
3. The authors built up on the original code base which only had the codes for the Prediction task with memory. I believe this is a significant achievement and hope that this experience was highly educational and rewarding for the authors. However, the original code repo has now been updated. So, it would be helpful if the authors could comment on discrepancies (if any) in their implementation and the one provided by the original authors in their final version.
4. The authors mentioned performing a hyperparameter search and reported finding different set of hyperparameters than originally reported in the paper. Having a plot of the performance variance for different hyperparameters (in the supplementary) could add value to this effort.
5. I would like to appreciate the several ablation studies performed by the authors to evaluate various components of the proposed methodology. The inferences from these studies though could be presented better.
6. The authors performed useful additional experiments beyond the ones performed in the original paper. These experiments definitely add to my understanding of the method. Although I could follow the results of these experiments, I was slightly confused about the inferences presented by the authors. However, since this is a reproducibility report, I gave more weightage to the rationale behind the experiments rather than their interpretations.
7. The authors allude to some reproducibility recommendations but it's not very explicit. I think these recommendations are extremely useful and it would help to have them clearly mentioned.


Despite the several merits of the report, I have few concerns primarily pertaining to the presentation and writing/organization style of the report. I have detailed them below.
__Cons:__
1. It would help to have a figure/flowchart that outlines the several experiments that the authors tried to replicate and/or investigate in detail with additional experiments. I would strongly suggest adding this figure in the introduction or scope section. This figure would establish the context and methodology of the original paper as well as give the reader an overview of the rest of the report.
2. With the original code repo being updated recently, the authors may have to briefly comment about the discrepancies (if any) in their implementation. Additionally, they need to update the report to incorporate the information that the original repo now contains the codes to both tasks. I don't think this needs to be very detailed but updating this information will add to the completeness of the report.
3. Although the authors report that they communicated with the original authors, it is slightly unclear as to how this communication served to improve the authors' understanding and/or implementation of the methodology. Also, I am curious to know if the authors communicated the observed discrepancy about the ShanghaiTech dataset and the engineering fix they had to deploy.
4. It would help to have a plot of the performance variation with changing hyperparameters. This result would indicate how sensitive the results are to hyperparameter values. Also, did the authors try different seeds? I understand this could be difficult to do on complicated dataset but it is an essential component to establish that one method statistically outperforms another method. As a recommendation, I would propose the authors to run their "light weight" experiments on multiple seeds and report the standard deviation in metrics.
5. I think the overall presentation style needs to be improved. The current report has very interesting experiments and excellent insights. These insights need to highlighted properly. For instance, it will be helpful to have the results reported by the original authors alongside the results obtained in this report to clearly compare and contrast the discrepancies or agreements. Furthermore, it would help improve the readability if the authors could add a flowchart or table that summarized their findings in terms of which modules turned out to be redundant for each of the tasks. Having such a figure or table will significantly improve the impact of their findings.
6. The discussions and implications presented by the authors don't seem very convincing. For instance, the authors mention _"Figure 9 shows both models having trouble reconstructing the input frames, but the Prediction model reconstructs the image with considerably more artifacts making it easier to classify correctly."_ It is hard to conclude this without definite reconstruction error (MSE, perhaps?) values.
7. Furthermore, did the authors try an epsilon-greedy type of approach to mitigate the problem of mode collapse in memory module? This would entail using the memory module as proposed with probability of (1-epsilon) and using a random combination of all memory items with epsilon probability. The value of epsilon could become another hyperparameter, but setting it to some constant value (eg. 0.1) could perhaps serve as an acceptable solution. However, this would be a minor comment and mostly out of my own curiosity. I have not used this concern to judge the merit of the reproducibility report.
8. Finally, I would encourage the authors to add clear reproducibility recommendations. I believe the authors are at an appropriate stage to provide specific recommendations for this work and a good reproducibility report should have them.

Once again, I would like to congratulate the authors on their work and I hope they can address some of the concerns mentioned above to improve the impact of this report. I believe that if the authors could add the aforementioned summary figures and/or tables and improve the overall readability of the final version of the report, this work could be extremely useful to the field in general.

**Familiar With The Original Paper:**

I have read the original paper

**Reproducibility Summary:**

Report has summary

---

### Official Review · AnonReviewer3 · 2021-03-11
**Strong yes.**

**Rating:** 9
**Confidence:** 3

**Review:**

The authors reproduced the work of Learning Memory-guided Normality for Anomaly Detection.

In this work, the authors have:
- Reused part of the original code and scripted some parts that were not available
- Have communicated with the authors regarding the changes
- Did a thorough ablation study

And finally, went beyond the scope of the reproduction and suggested further improvements.
In the reviewer's opinion it is a thoroughly conducted study.

Small typos: Space missing line 45


**Familiar With The Original Paper:**

I have read the original paper

**Reproducibility Summary:**

Report has summary

---

### Decision · Program_Chairs · 2021-03-31

**Decision:**

Accept

**Comment:**

Selected for ReScience-C Journal Publication.